# Kubism™: Picasso, Trademarks and Bouillon Cube

## Noam M. Elcott

Department of Art History & Archaeology Columbia University, New York, NY 10027, USA;
nme2106@columbia.edu

**Abstract:** Pablo Picasso's *Landscape with Billboards* (1912) evinces a deep and complex relationship with emergent trademark and related intellectual property law in France. Among the three trademarked logos featured prominently in the work is that for Bouillon Kub. Critics, caricaturists, and the Cubists themselves toyed with the visual and textual rhymes between Cubism and Bouillon Kub. But only Picasso in his *Landscape with Billboards* engaged deeply with the nascent trademark and design protection laws exploited more forcefully by Bouillon Kub than nearly any other brand. This essay is a small part of a larger chapter on Picasso, Cubism, and the semiotics of trademark, which, in turn, is a part of the book project *Art™: A History of Modern Art, Authenticity, and Trademarks*.

**Keywords:** Pablo Picasso; Cubism; advertising; trademark law; intellectual property law; design

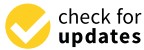



Picasso's early 1912 painting, *The Bouillon Cube*, and his more ambitious effort, *Landscape with Billboards* (Figure 1), painted in the summer of that year, prominently feature the trademarks and trade dress of KUB, the innovatively fabricated and aggressively advertised bouillon recently introduced to the French market by the Swiss-German industrial-food manufacturer Maggi. At first blush, Picasso's recourse to KUB can be viewed as part of the Cubist play with words in which "any word containing the sound 'cube' was immediately embraced," whether bouillon KUB, the Czech violinist Jan Kubelík, the Czech artist Bohumil Kubišta, or the Austrian printmaker Alfred Kubin.[1] Bouillon KUB, however, was plainly something more.

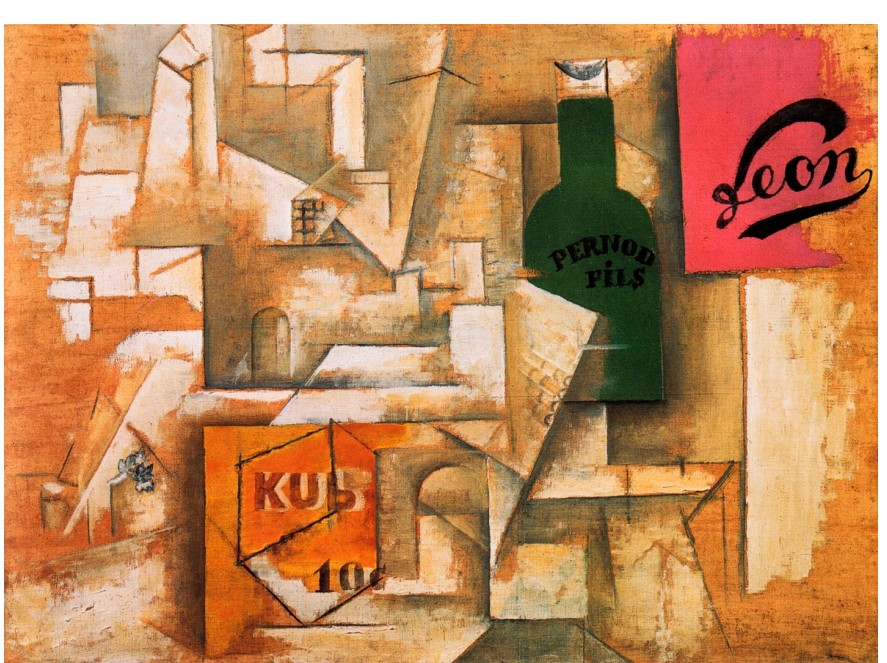

**Figure 1.** Pablo Picasso, *Landscape with Billboards*, 1912. Oil and Ripolin-brand enamel paint on canvas. 46 × 61 cm. The National Museum of Art, Osaka.



Bouillon KUB was not a genuinely new type of product—it received no patent protection—but rather a new form for an old standby. Maggi was long in the business of aromas, flavors, consommés, and bouillons, among other products, produced at industrial scale through closely guarded trade secrets.[2] In 1907, a German chemist named Graf attempted to register patents for the manufacture of bouillon using the precise methods employed by Maggi. With their trade secret disclosed, Maggi executives searched frantically for a new product, which they could protect more meticulously than their erstwhile trade secrets. The solution was a new form, the cube, for the familiar bouillon, secured through a range of intellectual property protections. First and foremost, the product required a distinctive trademark. Descriptive names—such as Coca-Cola for beverage infused with cocaine from the coca leaf and caffeine from the kola nut, thus: Coca-Cola—generally worked best in advertising. But if the potential mark was deemed necessary to describe the product—no one soft drink company can monopolize the term "cola"—registration would be challenged and likely refused, as was ultimately the case for Coca-Cola and its attempt to deny the registration of all other colas. Maggi initially attempted to register "cube" but was thwarted by the commercial court because, in the court's estimation, "['cube'] could become the necessary designation (*appellation necessaire*) of a product."[3] The court's reasoning was sound and widely accepted: If Maggi owned exclusive rights to the term "cube" in the context of bouillon, competitors could not reasonably enter the market for bouillon cubes. "KUB" proved an ingenious alternative: distant enough from "cube" to abide by the ruling, proximate enough to "cube" to be highly suggestive of the product. (As a rule, misspellings and phonetic equivalents were frowned upon, when not rejected. Maggi was pushing the envelope of the law.) Furthermore, Maggi made a virtue of the awkward spelling as endless advertisements, posters, and billboards entreated: "Demand the KUB with a K" or simply "Demand the K" (Figure 2). The manufacture processes could not be protected through patents as they were already in the public domain, but the intertwined name and form lent themselves to trademark and trade dress protections. Rather than sell mere bouillon, a highly competitive market, or even bouillon cubes, which would soon be a crowded field, Maggi sold KUB.

In order to secure KUB as its exclusive intellectual property, Maggi mobilized multiple arms of the law. On 26 July 1909, a Maggi subsidiary successfully registered the name and packaging of Bouillon KUB (Figure 3a). (A trademark registration in the name alone would be filed on 24 February 1910 (Figure 3b)). According to the 1909 trademark registration: "This mark consists of a box in cubic form, the four sides of which are surrounded by a label and the upper face also covered with a label."[4] More aspirational than technical, the registration sought to secure "*une boite de forme cubique*" as its trademark. By the time competitors like Oxo and Knorr followed with their own bouillon cubes, Maggi had already secured key aspects of the cube as its own intellectual property. Yet, it was hardly satisfied. In order to maximize its options in a fluid legal landscape, Maggi was among the first companies to shore up its packaging through the "Law of 14 July 1909 on Designs and Models."[5] On 26 July 1910, a Maggi subsidiary representative successfully deposited a model of a metal box for "alimentary products" designated by the name "Bouillon Kub" (Figure 3c). In addition to the now-familiar exhortations to "Demand expressly the trademark: Bouillon 'KUB,'" the vividly colored etiquette announces that "The veritable bouillon KUB is sold exclusively in a red package." To which was appended the phrase: "brevetés s.g.d.g.," an abbreviation for "brevetés sans garantie du gouvernement" or "patent without government guarantee," a now-defunct French patent registration that required no examination. However imposing the abbreviation appeared, it was likely meaningless for this product. Future advertisements built on the intellectual property filings to refine the visual training of consumers even further; they unequivocally instructed that KUB is sold "exclusively in red packages with yellow letters." Among the consumers that would heed these exhortations closely was Pablo Picasso. For while he was an unlikely purchaser of bouillon, he was undoubtedly an ardent visual consumer of advertisements for KUB.

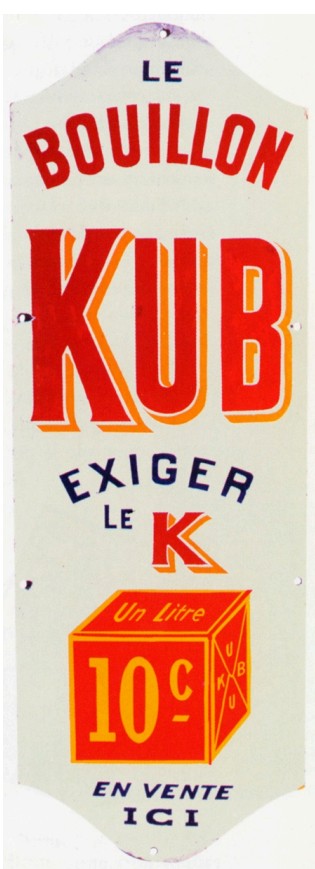

**Figure 2.** Enamel advertising sign for the launch of Bouillon Kub in France in 1908.

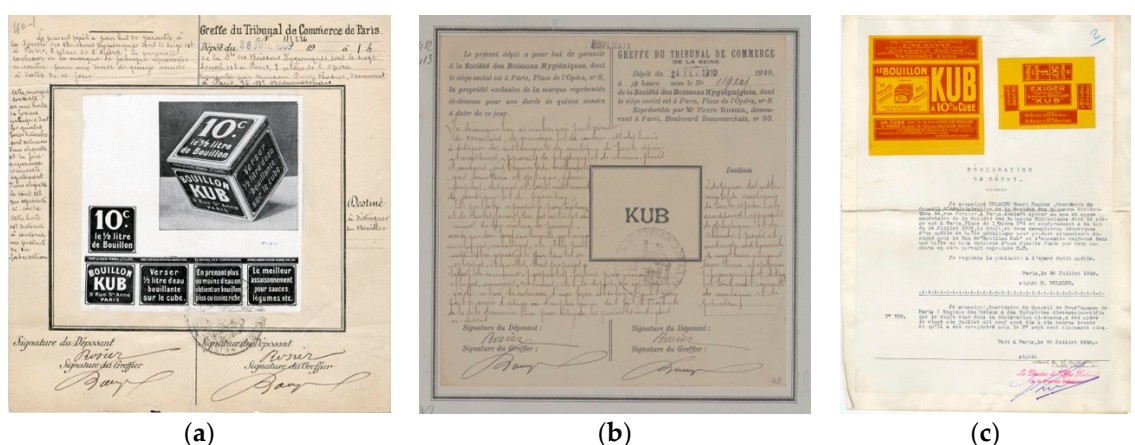

| (**a**) | (**b**) | (**c**) |

**Figure 3.** Left to right: (**a**) trademark registration for Bouillon KUB, consisting of "a box in cubic form," 26 July 1909; (**b**) trademark registration for the mark "KUB," 24 February 1910; (**c**) model registration of a metallic box for an alimentary product designated by the name "Bouillon Kub," 26 July 1910. Images (**a**,**c**) courtesy of Archives Historiques Nestlé, Vevey. Image (**b**) courtesy of the archives of the French Patent and Trademark Office (INPI), Paris.

An 1881 law on the freedom of the press enabled *afficheurs*, workers who posted bills on city walls, to post without obtaining prior authorization from the prefect of police. In short order, posters remade the visual topography of France's cities. As Ruth Iskin relates, posters were plastered over with new advertisements so quickly that advertisers turned to rented space on building walls, resulting in the commodification of public space and the transformation of facades into revenue producers for building owners, companies,

and municipalities.[6] In the years before the First World War, as Jeffrey Weiss chronicles, posters exploded in size to become *panneaux-réclame* or large billboards, and populated the landscape around Paris and other suburban environs. Referred to colloquially as "*les barre-la-vue*" (blocks the view) and "*les affiche-écrans*" (poster screens), these giant billboards became objects of civil scorn. In the summer of 1912, the French Finance Ministry imposed a prohibitive tax hike on the billboards, to the consternation of advertisers but with support from nearly all other parties.[7] It was precisely in these years of heated debate that the avant-garde found itself tangled up with advertisements and billboards, none more pointedly than bouillon KUB.

As recounted in 1911 by Guillaume Apollinaire, a principal champion of the avant-garde, one could not escape advertisements for Kub, nor their resonance with Cubism:

Nowadays, it is easy to joke about new works of art. This dispenses with the need to understand them. Today, we laugh at the Cubists. Yesterday we laughed at the great painter Henri Matisse. He himself gave in to this modern flaw. He laughs, they say, at Cubism; he came up with this name. How disappointed he must have been this summer when he arrived in Collioure, where he was spending the season, to read the word "Kub" on the house in enormous letters! The exterior of one of the walls of his house was rented to a publicity firm that advertised this trendy food item. Thus was played on the master of commanding and mellifluous colors a trick that spoiled his vacation.[8]

By 1912, seemingly all of Cubism's enemies marshalled bouillon Kub to ridicule the ascendant art movement. Louis Vauxcelles, among the first critics to dismiss the art of "Bracke" (sic! Georges Braque) as "*bizarreries cubiques*,"[9] now channeled the avant-garde's passion for *King Ubu* (Alfred Jarry's notorious 1896 absurdist drama) and Kub's omnipresent packaging (featuring the letters K-U-B-U in triangular quadrants) to coronate Picasso with a loathsome crown: "Picasso who, ten years ago, was not lacking in talent, is the leader of the Cubist gentlemen, something like Father Ubu-Kub."[10] The caricaturist Joseph Hémard similarly mashed up references to the avant-garde and marketing in a "Cubist" portrait of Armand Fallières, President of France, for the satirical magazine *Le Rire* (Figure 4). Along with Cubist mainstays such as a bottle and glass of wine, dice, and a crystalline, faceted surface, Hémard reworked KUB's popular slogan, "Demand the K," into a mock Cubist advertisement: "With Cube painting, demand the Cu."[11] Finally, it was Hemand's July 1912 caricature for *La Vie Parisienne*—in which a cubic and (nearly) two-dimensional painter paints a billboard advertising "Bouillon in 3 dimensions"[12]—that laid the groundwork for Picasso's contemporaneous rebuttal (Figure 5).

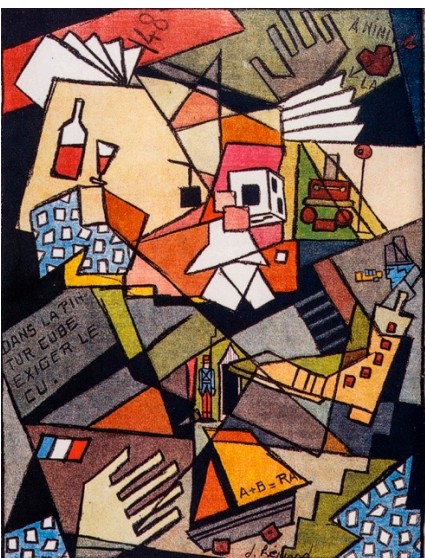

**Figure 4.** Joseph Hémard, *Portrait of Armand Fallières*, published in *Le Rire.* 26 October 1912.

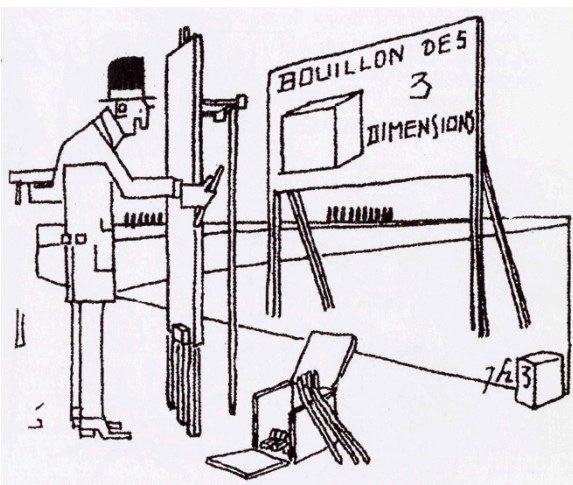

**Figure 5.** Joseph Hémard, *Master Cube on Vacation*, published in *La Vie Parisienne*. 13 July 1912.

What, under these circumstances, could Picasso bring to debates on modern advertising, let alone to the intricacies of established and emergent trademark laws? How could Picasso secure for himself the last laugh as Father Ubu-Kub? Picasso's 1912 *Landscape with Billboards* features the industrial landscape and giant billboards the artist would have encountered on the Sorgues-Avignon tramway, which he regularly rode while staying at Sorgues to escape from the pressures of Paris.[13] The landscape remains sketchy, and the painting survives in poor condition. But we know Picasso valued it as he wrote about repeatedly to Kahnweiler.[14] A fastidious visual account is imperative. To the far right, but at an uncertain distance, stands a bright pink billboard for Léon, a largely forgotten men's hat manufacturer. In the middle ground, an imposing, emerald green, bottle-shaped billboard for Pernod Fils, the reigning producer of absinthe, a producer and a drink well represented in Picasso's oeuvre. Finally, in the foreground, a rusty-yellow billboard surmounted by the awkwardly capitalized "KUb" in muddy-red lettering and the familiar 10¢ price. In Picasso's depicted and intellectual landscape, Léon and Pernod Fils are mere foils for the polemical deconstruction of the "Kubist" advertisement. The attack unfolds in two stages, each executed with brutal precision.

First, *Landscape with Billboards* was an attack on Salon "Cubism." Perverse as it may seem, around the summer of 1912, Picasso was hostile to "Cubism." A year earlier, a group of artists including La Fauconnier, Léger, Delaunay, Metzinger, and Gleizes, collectively identified as "Salon Cubists," exhibited at the Salon des Indépendants to great ridicule and acclaim.[15] Picasso and Braque, with the support and guidance of their dealer Daniel-Henry Kahnweiler,[16] tried to avoid all association with the Salon Cubists and largely refused to exhibit in Paris. As a result, Picasso and Braque commanded greater reverence and prices than their followers or, as Picasso described them, "the awful stragglers."[17] The disadvantage, of course, was that their works were so difficult to see in Paris that they ceded much of the public reception to Metzinger and company. By mid-1912, Metzinger and Gleizes claimed the mantle of invention and leadership of Cubism, an assertion soon to be buttressed by their widely anticipated exhibition, the "Salon de la Section d'or," and forthcoming treatise, *Du "Cubisme"* (both late 1912).[18] Rather than engage in a turf war over proprietary ownership of Cubism, Picasso, Braque, and Kahnweiler responded with tactical jujitsu: they abandoned "Cubism" to the arrivistes and advanced their own aesthetic away from a style rapidly descending into academic kitsch.[19] *Landscape with Billboards* was not a proclamation of Picasso as the rightful "Father Ubu-Kub," but the opposite. Picasso joined forces with the enemies of his enemies to attack Salon "Cubism." Rather than a mark of authenticity, the KUB billboard pronounces "Cubism" to be nothing more than a marketing gimmick, a brand. We can almost hear Picasso sneer: "For 'Cubist' painting, demand the KUB".

Second, *Landscape with Billboards* was a thorough deconstruction of trademarks, or rather, of a specific trademark: KUB. We can safely assume that Picasso knew nothing of French trademark law. Instead, he was a keen observer of contemporary advertising, which he studied as a predator scrutinizes its prey. The structural resemblance between Picasso's Cubism and bouillon KUB must have struck the artist as particularly potent: a reduction of the world to cubic forms; complex combinations of words and images, letters and shapes; abstract ideas rendered graphic; the exploitation of advanced means of communication; and the redistribution of received laws, whether timeworn or of recent vintage. KUB and Cubism shared more than a name and a shape. Both sought to remake the relationship between things and signs, images and words, form and color, authentic product and rivalrous imitators. Picasso did not dismiss trademarks as foreign and contemptable symbols, but rather engaged them as potent peers. The proximity seems to have fueled Picasso's predatory instinct, for he deconstructed the trademark with merciless precision.

As established in Maggi's various trademark registrations in France—and as advertised by the advertisements themselves—the KUB mark comprised a number of related elements, to which adhered varying degrees of legal protection. First and foremost was the name KUB. Maggi invested substantially in the promotion of the name KUB. The name almost always appeared in advertisements fully capitalized, KUB, though it was also often referred to as a regular proper noun, Kub. Picasso slyly splits the difference to compromise the integrity of the name: KUb. Second, the box in the form of a cube, registered as a two-dimension trademark and design as well as a three-dimension model. French courts and legal theorists remained ambivalent as to the trademark registrability of the shape of packages, such as bottles and boxes.[20] And just as they rejected the trademark "cube," they were unlikely to uphold the exclusive rights to the cubic shape. Although Maggi could hardly monopolize a fundamental shape—that would be the crowning, if precarious, achievement of even more ambitious trademark holders—it successfully made the cube a centerpiece of its marketing campaign.

Picasso's Cubism, by contrast, rarely featured cubes at this juncture. Picasso's brazen recourse to a Platonic cube was surely part of the polemic in *Landscape with Billboards*. Upon close inspection, the depicted cube is a bundle of contradictions. The cube itself is delineated inconclusively and alternates between transparent and solid states. Its highest vertex extends ever so slightly beyond the top edge of the billboard, rendering its spatial location paradoxical. The relationship between the "KUb" and "10¢" lettering, the depicted cube, and the billboard is entirely unresolved. Additionally, the whole ensemble quivers ambiguously behind, on, or before the earthen wall of a house with a rounded open door.[21] Much more than the cube (let alone KUB), such spatial and conceptual uncertainty engendered by colliding symbolic registers was the hallmark of Picasso's Cubism at this moment. If KUB's cube had to be sacrificed for Picasso's Cubism to materialize, so much the better. Finally, color. Nineteenth-century courts largely rejected trademark claims in color alone. But they recognized color as a contributory factor in infringement claims and as a potentially constitutive component of a trademark.[22] French law in particular endorsed the importance of color as a significant factor in trademarks.[23] It is no surprise, therefore, that numerous KUB advertisements announced that veritable bouillon KUB is sold exclusively in red packages with yellow letters. Picasso's rejoinder was almost too easy: he reversed the colors.

*Landscape with Billboards* is best understood as a polemic against the arrivistes. Were the integrity of the KUB mark fully dismantled, the attack on Salon "Cubism" would have been lost. But by the summer of 1912, Picasso had already begun his full-frontal assault on trademarks as such. He selected several of the most litigated trademarks in Europe and deconstructed them across scores of works in every medium in which he then experimented. The full history of Picasso, Cubism, and trademarks is yet to be written.[24] But when it is complete, there can be no doubt that "Cubism" will not be written with a "K".

**Funding:** This research received no external funding.

**Data Availability Statement:** Data are contained within the article.

**Conflicts of Interest:** The author declares no conflicts of interest.

## Notes

1    (Rabinow 2014, p. 173) See also (Apollinaire 1972, p. 258).

2    On the history of Maggi and its world-famous bouillon cube, see (Pivot 2002, especially 65–78)

3    Quoted in (Pivot 2002, p. 68) "*Appellation necessaire*" is a legal term of art addressed at length below. French courts had long established that terms such as *extractum carnis* or *of meat* were necessary and thus generic in relation to bouillon. See (Pouillet 1892, p. 75).

4    All trademark registrations were also published by the official bulletin of industrial and commercial property. See (BOPI 1909).

5    On the "Loi du 14 juillet 1909 sur les dessins et modèles," see (Pouillet et al. 1911) The law was written for two- and three-dimensional industrial designs, not for packaging per se. In the nearly 1000 pages of the treatise, there is no discussion of packaging. Maggi was again pushing the envelope of the law.

6    See (Iskin 2014, p. 18).

7    See (Weiss 1994, pp. 62–65) See also (Chessel 1998, pp. 145–80).

8    (Apollinaire 1911).

9    (Vauxcelles 1909, p. 2).

10   (Vauxcelles 1912, p. 2).

11   (Hémard 1912a) Note the pseudo-Germanic spelling of "Pintur": "Dans la Pintur Cube, exiger le cu." The Germanic associations of KUB and Cubism would come to haunt the movement (as well as Maggi) when France entered into WWI against Germany. See (Cox 2000, pp. 268–72; Pivot 2002, pp. 76–78; Richardson 1996, pp. 352–53).

12   (Hémard 1912b, p. 498).

13   See (Richardson 1996, p. 241).

14   See (Cousins 1989, pp. 397–98).

15   See (Robbins 1985, pp. 9–23).

16   Braque was the first to sign with Kahnweiler in 1912. Picasso waited until December of that year to sign a formal contract, ratifying their informal but potent working relationship. See (FitzGerald 1995, p. 34) On Kahnweiler's strategic distancing of Picasso and Braque from the Salon Cubists, see (Richardson 1996, pp. 207–19) Noteworthy as well is the fact that, at the end of 1912, Kahnweiler sent Picasso's small painting *Le Bouillon Kub* to the "Second Post-Impressionist Exhibition," curated by Roger Fry for the Grafton Galleries in London, an exhibition that included works by Picasso and Braque, but no other would-be Cubists.

17   (Richardson 1996).

18   The "Salon de la Section d'or" took place in October of 1912, around the publication date of *Du "Cubisme"*. (Gleizes and Metzinger 1912).

19   (Troy 2003, pp. 63–65).

20   See, for example, (Allart and Allart 1914, pp. 52–53; Coddington 1878, pp. 419–21, 32–33; Pouillet et al. 1912, pp. 62–67) Other jurisdictions were far less accommodating to the registration of the shape of packaging or products as trademarks.

21   Such "phenomenal transparency" would become a commonplace of Cubist criticism. See, for example, (Rowe and Slutzky 1963, p. 47).

22   For an overview, see (Browne 1898, pp. 274–79).

23   Among numerous examples, see (Barclay 1889, p. 120; Pelletier 1893, pp. 167–72; Pouillet et al. 1912, pp. 71–74).

24   I venture as much in the second chapter of my current book project: *Art™: Modern Art, Authenticity, and Trademarks.* For the "pre-history" of Picasso's extensive engagement with the world's "first" registered trademarks (those of Bass Ale), see (Elcott 2024, pp. 114–63).

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
