# Peer review of "Kubism™: Picasso, Trademarks and Bouillon Cube"

_arts, 1910_

Round 1

Reviewer 1 Report

Comments and Suggestions for Authors

This essay focuses on Picasso’s summer 1912 canvas Landscape with Billboards, which prominently features the logos “Leon,” “Pernod fils,” and “KUB,” displayed on billboards.  The author has done a great deal of research into the history of trademarks in general and of the “KUB” logo in particular.  Pages 4-5 trace the way that critics, in 1911-12, juxtaposed advertisements for the KUB brand with the contemporary work of Picasso and Braque.  However, the essay says surprisingly little about how the history of the KUB logo should affect our understanding of Cubism.

What’s missing is context.  There is no discussion of the role of trademarks in early 20th-century art or in Picasso’s Cubist work more broadly.  Nor is there any discussion of the formal changes that occurred in Picasso’s work in 1912, which might have cast more light on the role of the trademarks in this canvas. 

In terms of iconography, it would have been useful to compare Picasso’s 1912 Landscape with Billboards to Raoul Dufy’s 1906 Posters at Trouville, which reproduces a series of distinctive trademarks, many geometric in character.  Evidently, Dufy’s painting is more realistic than Picasso’s.  But how does this difference affect the visual effect and significance of the trademarks? 

Regarding trademarks in Cubism, it would have been useful to compare Picasso’s two treatments of the distinctive bottle of Anis del Mono.  The proto-Cubist faceted bottle appears without a label in the 1909 Still Life with Anis del Mono (MoMA); it is reprised with a label in his 1915 Bottle of Anis del Mono (Detroit Institute of Arts).  In the interim, Juan Gris created a Bottle of Anis (1914, Reina Sofia Museum), where the facets are painted but accompanied by an actual label, pasted to the canvas.  The co-existence of different modes of reference in this motif should make it into a locus classicus for the analysis of trademarks in Cubism. 

It would also have been useful to consider the distinctive triangular trademark of Bass Ale, which figures in several of Picasso’s Cubist works, for instance his 1912 still life, Violin and Bottle of Bass.  It is surprising that the author does not mention this, given that his essay cited in note 24 discusses Manet’s Bar at the Folies-Bergère, where the Bass label with a red triangle appears at lower left and lower right.  Perhaps the red triangle was particularly suitable for trademarking, in keeping with the legal logic described on pages 2-4.  Does this geometric motif assume a new meaning within the context of a Cubism? 

Turning to the “KUB” in Landscape with Billboards, the author describes Picasso’s treatment as a “deconstruction” of the trademark (lines 156 and 168).  As the author notes (lines 199-200), the colors are reversed from yellow letters on a red ground to red letters on yellow.  The cubic container shown from above in advertisements is represented from below.  “KUB” becomes “KUb” (as noted in lines 174-176).  This linguistic truncation is a familiar phenomenon in Picasso’s Cubist work.  For instance, “Bass” often becomes “Bas” (low).  But do these modifications of the KUB box and logo qualify as “deconstruction,” a term typically used to describe the exposure of a hidden ideology beneath a seemingly neutral representation?  They seem more like the results of unmotivated formal play. 

In lines 102-116, the author demonstrates that the popular press of 1912 was replete with sarcastic remarks and caricatures referring to “Kubism.”  The emphasis on the German “K” relates to the frequent criticism of Cubism as a “Germanic” style, reflecting its popularity with German collectors.  This might have been explored in greater depth. 

At line 135 (and again at 202), the author suggests that Landscape with Billboards was fundamentally intended as an attack on Salon Cubism (lines 135 and 202).  This seems like a non-sequitur, unrelated either to the popular critique of “Kubism” or to the idea of “deconstruction.”  In any case, to demonstrate that Landscape with Billboards is an attack on Salon Cubism would require a formal analysis of the way that Picasso’s canvas does or does not resemble contemporary landscape paintings by painters such as Albert Gleizes and Jean Metzinger.  Without such an analysis, it is impossible to evaluate the proposed interpretation. 

It would also be useful to situate Landscape with Billboards within the evolution of Picasso’s own work.  To some extent, the 1912 canvas seems like a reprise of his summer 1909 Factory at Horta de Ebro (Hermitage): note in particular the chimney at upper left.  However, the geometric planes no longer enclose solid, three-dimensional forms, as they do in the earlier canvas.  The overall simplification of form may indicate that the painting is unfinished, or it may be a transition to the minimal architecture of Picasso’s fall 1912 papiers collés. 

The strong colors of the billboards recall the passages of blue, pink and yellow that Picasso introduced in his paintings of March 1912, painted with Ripolin enamel.  Picasso’s use of a commercial paint like Ripolin was noted by his contemporaries, who saw it as an assault on the sanctity of oil painting.  Do the billboard colors take part in this assault?  Unfortunately, the caption for this image is missing (as are the captions for all the other images).  Traditionally, the medium for Landscape with Billboards has been described simply as “oil on canvas,” but perhaps an up-to-date caption from the National Museum of Art, Osaka, would indicate the presence or absence of enamel paint.  Picasso seems to be endorsing the presence of billboards within the landscape, which he may have seen as comparable to his and Braque’s insertion of “printed” lettering into their paintings of 1911-12 (and to his utilization of collage in his March 1912 Still Life with Chair Caning). 

However, there is something anomalous about the billboards.  Trademarks such as the “Bass” label typically perform the same function in Cubist painting as the “real details” discussed by Kahnweiler and other critics.  They allow viewers to recognize a bottle in the same way that f-shaped sound holes or a carved scroll allow them to recognize a violin.  Similarly, as the author notes at line 191, the curved door at the left of Landscape with Billboards identifies the horizontal rectangle to which it is attached as a house.  The trademarks on the billboards, “Leon,” “Pernod fils,” and “KUB,” conspicuously fail to perform this function.  The presence of these free-floating signifiers imbues the painting with a proto-postmodern quality that seems worth remarking.    

Author Response

Many thanks for these thorough and thoughtful comments. Broadly, it seems that Reader 1 is asking for a full history of Picasso, Cubism, and trademarks. My excessively brief response is merely to rehearse what I wrote at the end of the essay and in FN 24: 

The full history of Picasso, Cubism, and trademarks is yet to be written. I venture as much in the second chapter of my current book project: Art™: Modern Art, Authenticity, and Trademarks. For the “pre-history” of Picasso’s extensive engagement with the world’s “first” registered trademarks (those of Bass Ale), see Noam M. Elcott, "The Manufacturer’s Signature: Trademarks and Other Signs of Authenticity on Manet’s Bar at the Folies-Bergère," Grey Room 94 (Winter 2024): 114-63. [NB. This essay should be available online and in print by the end of January.] 

In the much longer chapter, of which this essay is but one section, I address most of the issues raised by reader 1. In particular, Bass Ale figures centrally.  (Again, I chronicle the 19th century history at length in my forthcoming Grey Room article.) Indeed, I have identified around 60 instances where Picasso deconstructs the Bass logo and recently presented some of that research as a symposium at the Met Museum in conjunction with the exhibition on Cubism and the Trompe l'oeil Tradition. I also explore many other trademarks used and abused by Picasso. (Anis del Mono is just one of many, as reader 1 surely knows.) 

Reader 1 is wise to question my use of "deconstruction" and offers "unmotivated formal play" as an alternative. Given the brevity of the current article, this is a fair critique. But as part of a larger, systematic program to question and challenge proprietary ownership of signs (which is exactly what Picasso carries out in 1912-14), deconstruction is not too strong a word. I believe that hints of that deconstructive effort are visible already in Landscape with Billboards (and the other two readers appear to have sensed it). Additionally, they recognized that the attack on the arrivistes was not stylistic, but reducible to the moniker "cubist/cubism." Accordingly, I see no need to delve into the stylistic analysis called for by reader 1. 

Regarding the Ripolin paint, I will add a reference to Stark, Trevor. Total Expansion of the Letter: Avant-Garde Art and Language after Mallarmé. Cambridge, MA: MIT Press, 2020: p. 150-1. (though here "Pernod Fils" is misread as "Pernod Pils" and the color is misunderstood; it is in fact both mimetic of absinthe and emblematic of the advertised product, but unlike the colors of KUB, the green of Pernod is motivated by the color of absinthe and thus not available for any kind of trademark or other IP protection). 

Regarding the absence of captions: they will of course be added before the article goes to press. (Landscape with Billboards indeed uses Ripolin paint and I will be sure to include it in the caption.) 

Above all, I was gratified to see the seriousness of engagement. I hope that the longer chapter will live up to the high expectations of reader 1. Please do be in touch if you are interested to continue this conversation in a more direct manner. 

Reviewer 2 Report

Comments and Suggestions for Authors

This is a delightful and insightful paper retracing an episode where Picasso drew upon as well as critically engaged with contemporary forms of branding in the early 20th century.

Using the KUB trade mark (two words across much of the world, one word in America) as a vehicle, the analysis retraces the story behind Picasso's use of a Maggi stock cube or bouillon in his Landscape with Billboards.

The main target seems to have been an upstart wave of cubists who were using the label as little more than a marketing gimmick or brand.

Picasso was also deconstructing - and thereby challenging - certain exclusive claims to colour and form in the specific set of claims around the bouillon cube.

This is a rich and interesting episode at the intersections of art history, branding history and positive trademark law. However there is a broader legal literature looking at how advertising practices influenced trademark law across this period (see for e.g.  J. Mercer, “A Mark of Distinction: Branding and Trade Mark Law in the UK from the 1860s” (2010) 52(1) Business History, 17; Bartholomew, Mark. "Advertising and the transformation of trademark law." NML Rev. 38 (2008): 1) There is also a wider literature exploring the interface between art and trademarks more generally (when can an artists reuse a mark in a painting? when can a painting be repackaged as a trademark and registered indefinitely). The point is simply that art and trademark law have had a long history of encounters and would be nice to see that alluded to, at least in a footnote beyond FN22.

But these are offered as optional suggestions - it's an interesting paper and ready to go.

Author Response

Many thanks for these thoughtful and supportive comments. The Bartholomew reference is new to me and I will read it. The literature is indeed vast. I cite copious amounts of it in my forthcoming Grey Room article (which will be published before this article) and even more of it in the book. Due to space limitations, I kept references to a minimum in this essay. Given your obvious expertise in these matters, I hope we can be in touch. 

Reviewer 3 Report

Comments and Suggestions for Authors

Short article but well-written, easy-to-follow tantalizing content; excellent presentation of fascinating, rather new & important little-known aspect of Picasso's work; I hope you will be the one to write the "full history of Picasso, Cubism & trademarks!

Author Response

Many thanks for these supportive comments. I indeed hope to write a full history of Picasso, Cubism, and trademarks! 

Round 2

Reviewer 1 Report

Comments and Suggestions for Authors

In my response to the first draft of this essay, I noted that the author had done a great deal of research into the history of trademarks in general and of the “KUB” logo in particular.  Pages 4-5 traced the way that critics, in 1911-12, juxtaposed advertisements for the KUB brand with the contemporary work of Picasso and Braque.  However, the information in the essay lacked context.  There was no discussion of the role of trademarks in early 20th-century art or in Picasso’s Cubist work more broadly.  Nor was there any discussion of the formal changes that occurred in Picasso’s work in 1912, which might have cast more light on the role of the trademarks in this Landscape with Billboards.  I also noted that the illustrations lacked captions.

As far as I can tell, the only change in the new draft is that the illustrations now have captions.  These are useful.  However, in the first caption, it is not clear whether the information that Landscape with Billboards contains Ripolin enamel comes from the National Museum of Art, Osaka, or whether it simply incorporates my suggestion that it may contain enamel.

As I noted in my response to the first draft, the author demonstrates in lines 102-116 that the popular press of 1912 was replete with sarcastic remarks and caricatures referring to “Kubism.”  He then proceeds to the assertion, in line 135 that Landscape with Billboards was intended as an attack on Salon Cubism.  This remains a non-sequitur. 

Indeed, it is probably mistaken.  The critics’ emphasis on the “K” in KUB relates to the frequent characterization of Cubism as a “Germanic” style, which reflected the fact that the artists represented by art dealer D.-H. Kahnweiler—Picasso, Braque, Gris, and Léger—were exhibited primarily in Germany and that their work was purchased primarily by German collectors.  This was not true of the Salon Cubists, so there is no reason to see the satirical commentary as addressing Gleizes, Metzinger, et al.  Thus, if Landscape with Billboards is a response to caricatures like those of Joseph Hémard, it probably represents Picasso’s response to what he recognized as criticism of his own work.  Picasso typically responded to criticism by doubling down on whatever had offended his critic.  It thus seems likely that he inserted the KUB billboard into his summer 1912 canvas specifically as a defiant response to the cartoon by Hémard reproduced in figure 5.  The Salon Cubists are irrelevant to this visual exchange. 

Despite the unconvincing argument of the essay, it remains worth publishing.  The author’s meticulous research into the history of the “KUB” trademark provides valuable material for scholars of Cubism, and the possible dialogue between Hémard and Picasso constitutes an interesting episode for future analysis.